# Information System and Technology Optimization as a Tool for Ensuring the Competitiveness of a Railway Undertaking—Case Study

**Juraj Čamaj \*** , **Eva Brumerčíková** and **Michal Petr Hranický**

Department of Railway Transport, Faculty of Operation and Economics of Transport and Communications, University of Zilina, 010 26 Žilina, Slovakia; eva.brumercikova@fpedas.uniza.sk (E.B.); michal.petr.hranicky@fpedas.uniza.sk (M.P.H.)
\* Correspondence: juraj.camaj@fpedas.uniza.sk

**Abstract:** Information and communication technologies are becoming an increasingly important part of everyday life, as they facilitate many activities, mainly in the world of work, but also in scientific research and education. At present, informatics is one of the fastest growing sectors of the national economy. This development has had a significant impact on improving the quality of transport and transportation processes. The article is focused on the railway transport. It deals with the possibilities of planning the shifts of the train personnel and circulation of the vehicles. It describes the background of the topic. The scientific acquittance lies on the methodology proposed by authors. It presents a new idea of creating the shifts and circulations while being based on the current state and mathematical methods.

**Keywords:** railway transport; planning; shift; circulation

## 1. Introduction

Maintaining competitiveness in a liberalized rail transport market pushes the carriers to reduce costs when operating passenger trains [1,2]. One area where there is room for carriers to reduce costs is the area of better use of train requisites, i.e., traction rail vehicles, train drivers, train crews [2]. This can also be achieved by improving information systems or creating new ones.

Information and communication technologies are now becoming an increasingly important part of everyday life, as they facilitate many activities, mainly in the world of work, but also in scientific research and education. In practice, we can meet with information systems most often as their users, less often than their designers and planners. At present, informatics is one of the fastest growing sectors of the national economy. This development has had a significant impact on improving the quality of transport and transportation processes, because the basic pillars of the development of information technology are the demands to increase operational safety and streamline management and control activities in companies [3]. In rail transport, already existing information systems are being improved gradually. The trend is to use information systems that can communicate and share and use information and data from other systems [4,5]. In addition to the information processing itself, an important part of modern information systems is the storing and analyzing of information. Information systems and technologies are one of the tools for ensuring the competitiveness of companies [6,7]. In the field of railway transport, information systems are mainly used to optimize the use of means of transport and manpower, streamline operations, to increase safety and productivity, and to reduce costs [8]. From the point of view of the management of transport and transportation processes in railway operations, it is necessary for managers and designers to set operational requirements for information

and communication systems, which is a necessary prerequisite for knowledge of operational processes and information flows [9,10].

## 2. Background

Shiftplans are created directly by the carrier for its operational employees, who are most often locomotive drivers, conductors, stewards, etc. Individual job positions differ from each other mainly in the content of work [11]. In the area of train requisites, it is important to divide this staff into locomotive and train staff. Drivers belonging to the locomotive staff drive railway vehicles, not only on the trains that run according to the timetable, but also, for example, while moving the cars from the depot to the platform so that the set can then be dispatched as a train [8]. Conductors and stewards are train staff, sometimes referred to as accompanying staff. The role of these employees is primarily to be in direct contact with passengers, to sell and check travel documents, to provide information and assistance. There are certain differences between locomotive and train staff that affect the creation of shifts, so it is necessary to create shifts for these two types of employees separately [12,13].

In general, the carrier plays the role of an employer and the issue of shift of operational employees is focused on ensuring the operation of passenger trains by finding the optimal solution between the requirements of the employer (carrier) and employees (locomotive and train staff) [14]. The contrast between the requirements of employees and the requirements of the carrier can generally be characterized by two basic indicators. The basic requirements of employees are:

- Minimizing of workload,
- Maximizing the financial reward for the work performed [8].

In the case of workload, in terms of time, it is the total of time spent at work, which can then be analyzed as the time spent by the employee in the actual performance of work and the degree of his physical and mental workload in performing this work. On the other hand, the carrier (employer) has two basic requirements for its employees, namely:

- Maximizing the workload of its employees,
- Minimizing costs for your employees [8,15].

From the aforementioned, it can be concluded that the requirements of the carrier are antagonistic to the requirements of its employees, which is shown in Figure 1. It follows the necessary need to seek consensus in meeting these requirements so that the result is satisfactory for both employees and employers. It is, therefore, a matter of finding the optimal solution in terms of finding a match between workload and financial evaluation [16].

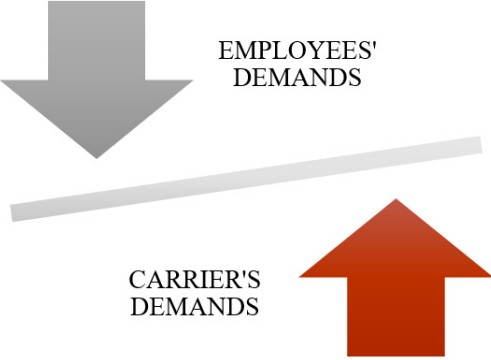

**Figure 1.** Representation of the conflicting requirements of employees and the carrier.

In a deeper analysis of the issue, it is possible to find a match between the conflicting requirements in case an optimal solution is found—meaning acceptable to both parties [17]. The optimal solution is to achieve a balanced ratio between the workload of employees and their financial evaluation for the

work performed [15,18,19]. The search for the optimal solution between the requirements of employees and the carrier is shown in Figure 2.

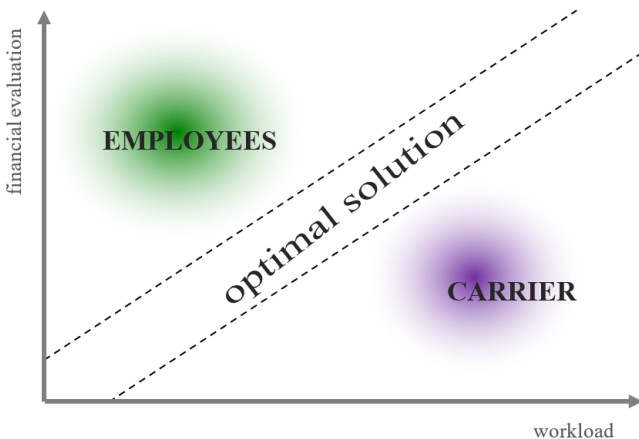

**Figure 2.** Representation of the position of the optimal solution between the requirements.

*Compilation of Shifts and Shifplans of Operational Employees in Information System KASO*

The KASO program, created by the Oltis Group, checks the proposed shifts, identifies their possible shortcomings, enables the sharing of all shifts within the company's internal network, operatively modifies individual shifts, and publishes them in predefined templates.

KASO is a database and modular system. The KASO program is used in the conditions of the carrier in a production environment (or in practical operation). IS KASO is an online system and is on the server, so all saved changes will be reflected in it immediately and immediately visible to all users who have access to the server. Modularity also provides individual tasks that are assigned to individual users, allowing access only to a selected module precisely determined according to the job position performed. The user neither sees nor can influence other modules [18,20]. The aim of the introduction of the KASO program in a transport company was to speed up and clarify both the carrier's processes in creating timetables as well as between ZSSK and ŽSR when ordering train paths.

KASO helps to identify deficiencies and potential errors in the created circuits and shifts of all train requirements in the conditions of ZSSK. However, the next step, such as proposals for optimization, is no longer the competence of this information system, but again only the employees who make and implement any corrections and adjustments in the information system. Optimization methods are not used in this information system [18]. It is a comprehensive tool for creating and checking correctness, but always taking into account the personal experience of employees.

Therefore, it is desirable not only to validate shiftplans, but also to think about the possibilities of optimization using all available scientific methods when designing them.

When compiling shiftplans, emphasis must be placed on the economy and clarity of processed [21]. The working hours of train drivers and accompanying staff result from approved shifts and are determined on the basis of a collective agreement, valid labor laws, regulations, and directives [19,22]. The shift of drivers and accompanying staff consists of work shifts and day offs. Work shifts and free days must be arranged in accordance with the shift rules in connection with the observance of a continuous rest between two shifts and a continuous rest during the week [22]. More than one work shift can occur within one calendar day (but usually a maximum of two). In connection with the creation of the most efficient and economical shifplans of drivers and accompanying staff, it is necessary to choose the most economically and practically advantageous interruption of the shift (readiness, interruption of shift, or interruption of shift with rest with the possibility of sleeping in bed) [23]. In case it is not possible to provide rest with bedtime or an emergency room, the economic advantage of individual interruptions and the use of the overhead route should be assessed [9,23]. The interface of KASO is shown in Figure 3.

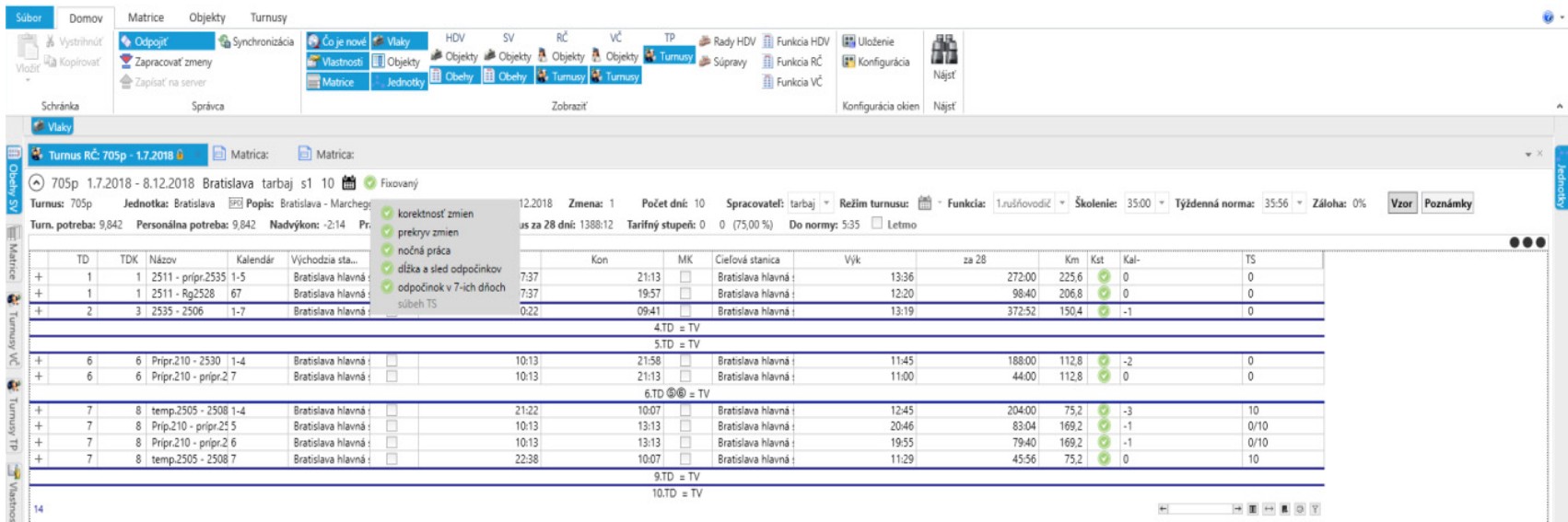

**Figure 3.** Checking the shiftplan of train drivers in IS KASO.

When compiling shiftplans of train drivers, IS KASO modules are used, which is conditioned by fixing the locomotive circulation. Driver's shifts are planned in connection with the circulation of rail vehicles. The driver's shiftplan must not be processed in a way that the number of turning days contained in it is divisible by seven. The accompanying driver's destinations (i.e., the destinations or trains on which the relevant drivers operate the service regardless of the home locomotive depot) are assigned to the operator by the carrier's management apparatus. The working time of a driver is calculated for a period of 28 calendar days on the form "calculation of working time for a shiftplan". The form is incorporated in the IS KASO module and serves as a document for calculating the working hours of drivers included in the relevant shiftplan [24]. Each calculation is performed only for the whole group of drivers or for one workplace, for drivers regardless of the locomotive groups. The chosen employee who deals with creating the shifplans is responsible for filling in the form correctly. They must have these calculations available during the approval of driver' shiftplan. They are also responsible for the correct completion of the "driver standard" form. The shiftplan should be published before its final approval for an approval by the employees' representatives [25]. Any conditions on the part of the employees' representatives will be worked out by the responsible employee and these re-approvals will be agreed on and sent in at least two copies for approval by the carrier's management apparatus [26].

For all train shiftplans that do not coincide with the circulation of rolling stock, an appropriate explanation must be provided in the footnote note. Individual days on which the driver does not perform the service according to the shift are referred to by the abbreviation "TV" (shift leave or day off). Performances that are not included in any shiftplan are considered to be "fleeting" shifts (unclassified performances).

Based on the order of the number of the accompanying staff, routes, and train restrictions by the carrier's management apparatus, the chosen carrier's employees will prepare a proposal to accompany the trains [27]. The proposed services will be redistributed to individual centers of operation. When allocating the proposed performances or individual shiftplans, it is necessary to ensure the escort of all required passenger trains with maximum use of working hours, provided that the productivity of shifts is increased [26,28]. If the assigned performances (or other requirements) of accompanying trains result in the need for incorporating the overhead route from the employer's point of view during the processing of the accompanying staff shift, the following aspects are taken into account: Shorter time when the driver is not controlling the locomotive (using the train as a passenger, public transport, walking) [29]. When it is not possible to determine a suitable train connection and there is a public transport connection in the required direction, the economic advantage of its use is assessed over the rest with the sleep on the bed.

When creating shiftplans of accompanying staff in IS KASO, the responsible employee assigns accompanying staff to assigned trains and checks objects on trains from the timetable database, which is taken from the infrastructure manager. In one of the modules in the IS KASO, the planner creates shifts, compiles the shiftplans of the accompanying staff, and checks the correctness of the shift with the shift schedule. Due to the impossibility of incorporating all alternatives resulting from the restriction of trains, the planner works in cooperation with the team leader and the commanders or other employees to accompany trains during holidays, construction works, and other unusual events of a longer-term nature [30,31].

If the shifts of the accompanying staff are not approved by the employees' representatives and the tour operator has not violated the provisions of the shift rules, the Labor Code, and the collective agreement, then all changes in the processed shift will be taken over immediately after approval by the carrier's management. The calculation of the working time for the accompanying staff's turn, the outputs of the tables for the accompanying staff and the title page of the tour are an inseparable part of the shiftplan [32]. An example of a shiftplan is presented in Figure 4.

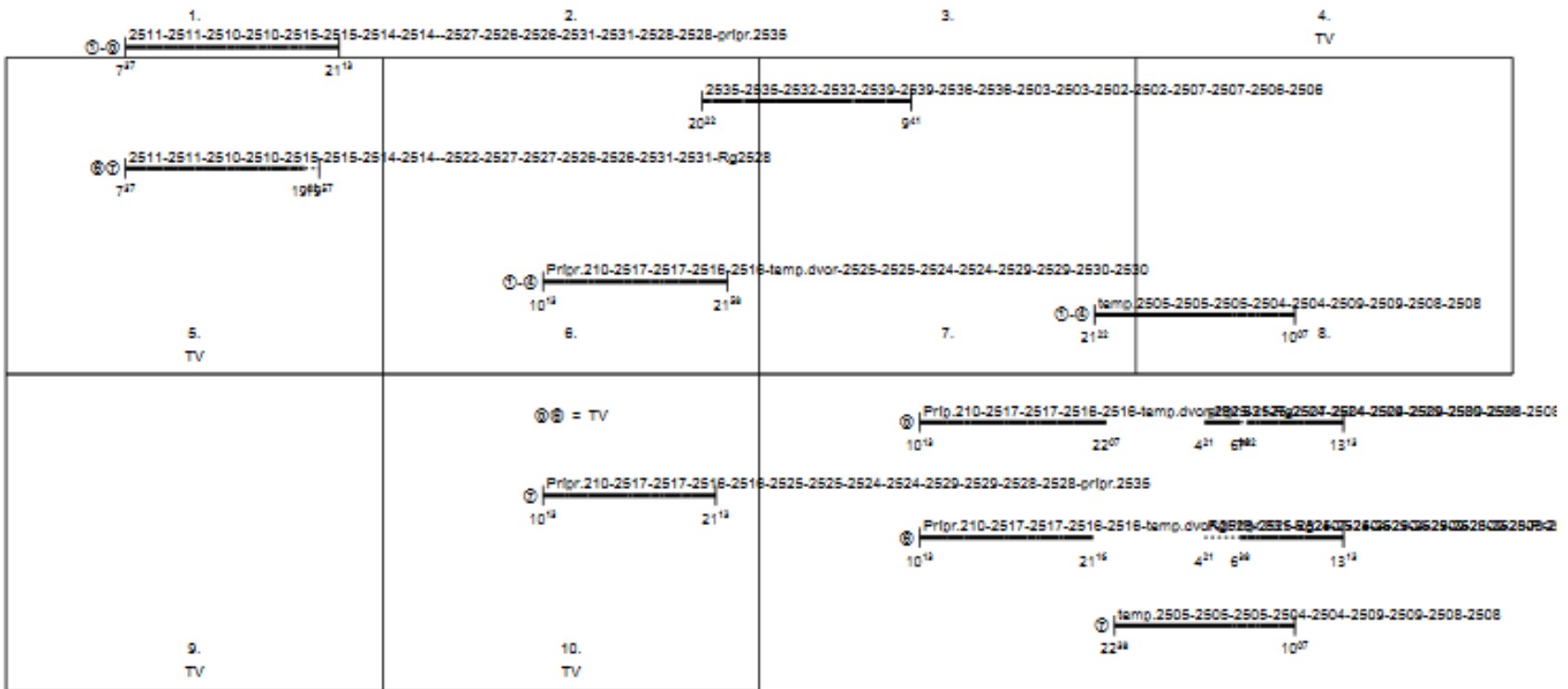

**Figure 4.** Shiftplan of train drivers—exit from IS KASO.

As for the place of start and end of the work shift, the inspection verifies the place of the beginning and the end of the work shift. If the start or the end station does not match the station of the home unit of the shift, the work shift is marked as incorrect. Continuity of performance: The control is used in all calculations. If the destination station does not agree with the starting station of the next performance, incorrect continuity of performances is indicated [33,34].

Checking the length of the work shift: The check verifies the length of the shift. The shifttime is not correct if the time from boarding to the arrival of the last train is longer than 12 h (for locomotive staff). In the case of a change with a rest for the duration of the previous performance, a condition applies that the rest, after a "short" first part of the shift lasting 3–6 h, must be equal to or longer than this part of the shift. The minimum rest period must be 3 h [34,35].

Minimum duration of work shift: If the length of work shift is shorter than the weekly standard —7 h, then the shift is marked as incorrect. Length of rest in a work shift: The check verifies the length of the rest in a shift [36]. A work shift is marked as incorrect if the rest is longer than 540 min. A shorter rest is correct. The amount of 540 min is bound only in the case of rest in a station other than the home station. Track performance: The check verifies the time from boarding to the last arrival. A work shift (or part of the shift) is marked as incorrect if the time from the start to the last arrival is longer than 12 h. Selected technology trains at the end of the controlled part of the shift are not counted [37].

Overlap of work shifts: The control monitors the overlap of shift on individual days of the shift. If there is a situation that in one day there are shifts that have an intersection in calendars and in time, including parts of shift from previous days, a violation of correctness is indicated. The control is involved in both weekly and daily mode [38,39].

Rest for seven days: The control guards a continuous rest of 48 h between shifts for seven consecutive calendar days. The check is performed only if the shift has passed the check of overlapping shifts. If there is only one sequence of shifts in the plan that does not meet this condition, then the shiftplan is marked as incorrect. If the continuous rest between shifts in seven consecutive days is at least 32 h, then the shift is marked as correct with the condition. In the event of a breach of the overlapping shift control, the continuous rest control is not performed. The control is involved in both weekly and daily mode.

Night work: The control guards the condition of "night" shifts on consecutive nights. A night shift is considered to be a change with a continuous output of at least 180 min (without interruption) in the time from 22:00 to 6:00. Length and sequence of rest: The check verifies the length of rest between the shifts. A shiftplan is marked as incorrect if the rest between shifts is shorter than 360 min. If the rest between shifts is shorter than 480 min, the shift is correct with a condition.

## 3. Results

Given that there is a large number of technological activities that ensure the operation of trains and some of them are interconnected and dependent, for better clarity, we analyze them according to four perspectives, with regard to:

- Locomotive driver,
- Accompanying staff,
- Passenger wagons,
- Locomotive.

This division is especially suitable for the automation of circulation, which are designed for these four categories. In some cases (complete units or push-pull sets), the last two categories can be considered as one [40,41].

Vehicle circulation and staff shifts include, in addition to the train running time itself, other necessary technological processes. At present (if they are taken into account) they are represented by an empirically given rate, which, however, may not correspond to the facts in many cases. These times vary depending on a large number of factors that we understand as input variables. Taking all factors

into account is currently very demanding due to the manual processing of circuits and shifts with some support from the KASO program. At present, the creation of circuits and shifts depends on the knowhow of employees and their many years of experience, because the KASO program does not design either circulation or shifts but is an excellent control tool [42,43].

If we consider the automatic machine creation of train vehicle circuits and train essentials, it is necessary for a possible program to be able to calculate the times determined for the necessary technological procedures. As these depend on a number of factors and vary considerably from case to case depending on several variables (number of wagons, type of the train, local conditions, etc.), the influence diagrams below present the procedure by which the system could select the correct combination of technological steps and calculate the required time. The inflation diagram makes it possible to assess the interconnectedness of the activities performed, but also their time sequence.

We understand the above variables as input parameters. The carrier can decide for itself which times are negligible and express them as zero. For the purposes of the model example, a specific procedure can also be expressed in the form of Gantt chart, but given that the individual times are not standardized, this expression would be quite inaccurate. In the influence diagrams below, each oval element represents the point in time at the beginning and the end of a certain necessary technological procedure that could be shortened in software processing. Each dark rectangle means a partial operation that lasts a certain time, and light blue diamonds are decision tasks. For a possible calculation, it is necessary to proceed in accordance with the diagram and to add up the times of partial operations [44,45].

The first example is in Figure 5, where the diagram shows all the necessary actions from the driver's arrival for a shift (this means the start of working hours, the shift figure from when the driver is paid) to the departure of the first train. The diagram in Figure 6 represents the procedure to be performed by the accompanying staff (conductor) when turning the train at the terminal (another train goes in a different direction). The diagram takes into account all possible variants of procedures (different types of vehicles, different type of station, availability of other personnel, etc.).

The first two diagrams were about the locomotive drivers and conductors. The last-chosen example from all the diagrams describes the technological processes with a set of wagons. Given that the only limiting parameter for calculating the optimal train circulation is the shortest time from the arrival of the train to the departure of the next train (it is pointless to calculate the time from arrival to decommissioning or from other technological processes to departure, as decommissioning should be limited as much as possible), below in Figure 7 we present an influence diagram for the turnover of a classic wagon set, which considers a combination of wagons of common construction with the locomotive.

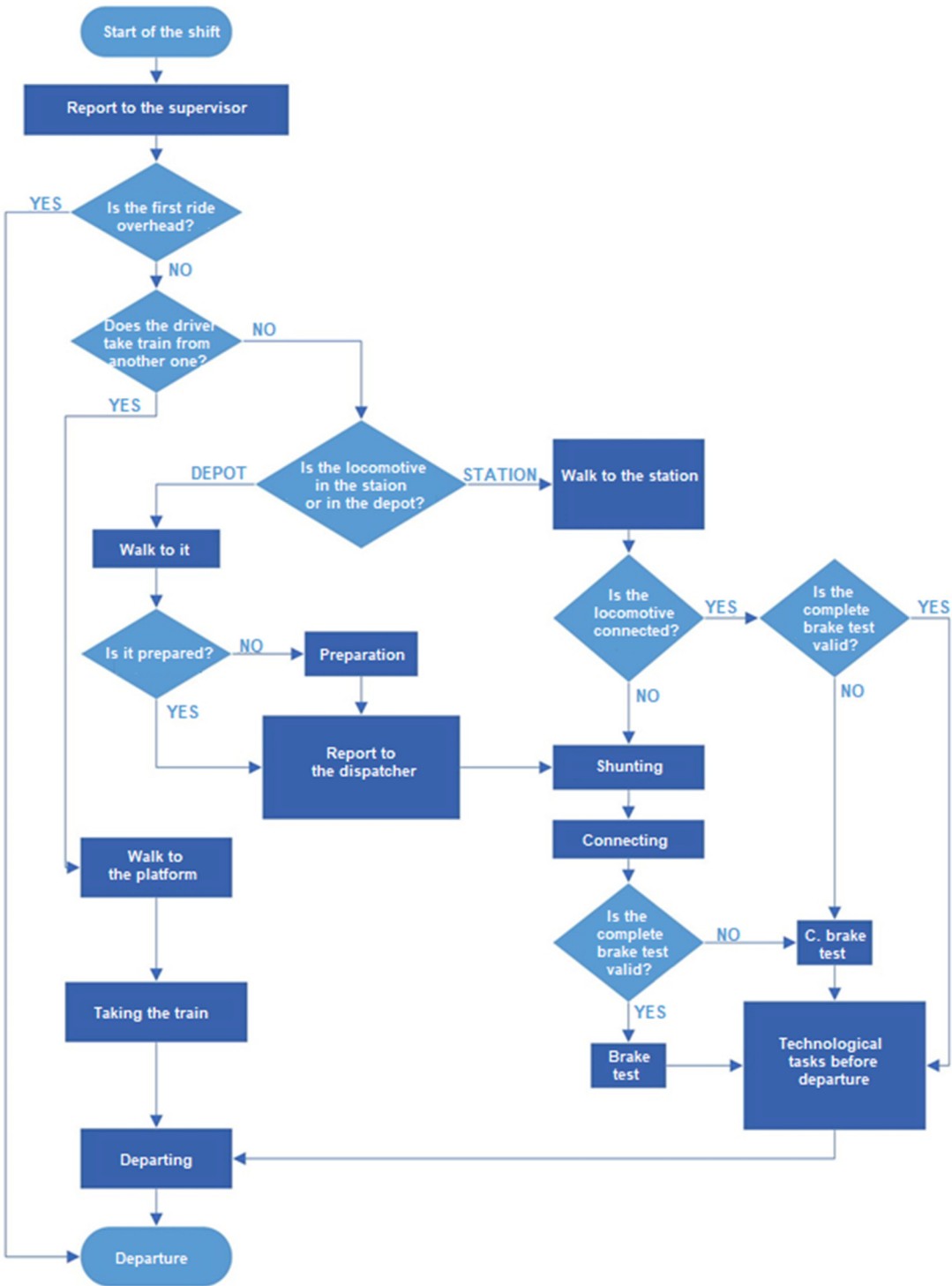

**Figure 5.** Influence diagram of technological operations from the arrival of the driver for a shift to the departure of the train.

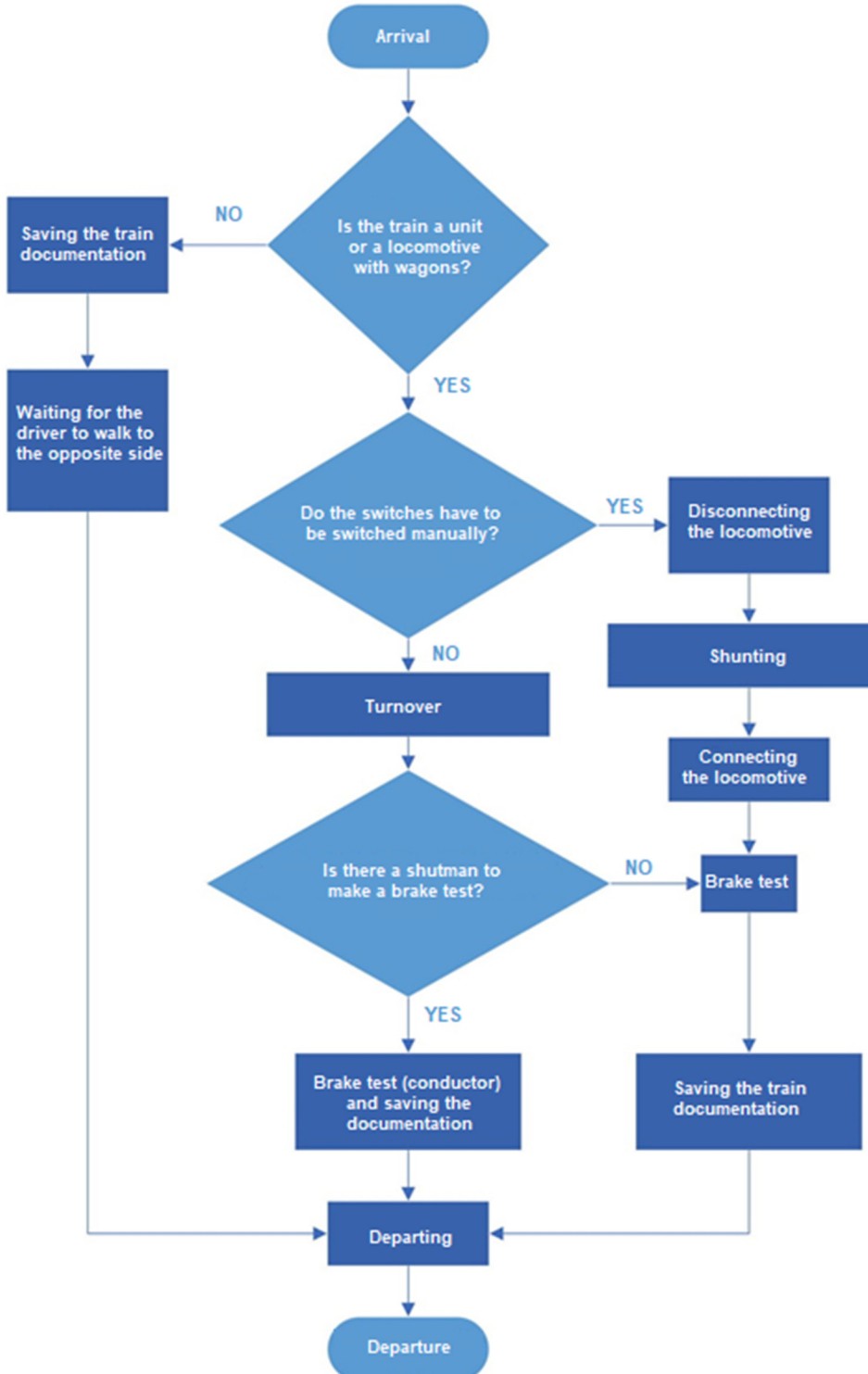

**Figure 6.** Technological operations of accompanying staff from the arrival of one train to the departure of another in the event of a turn at the terminal.

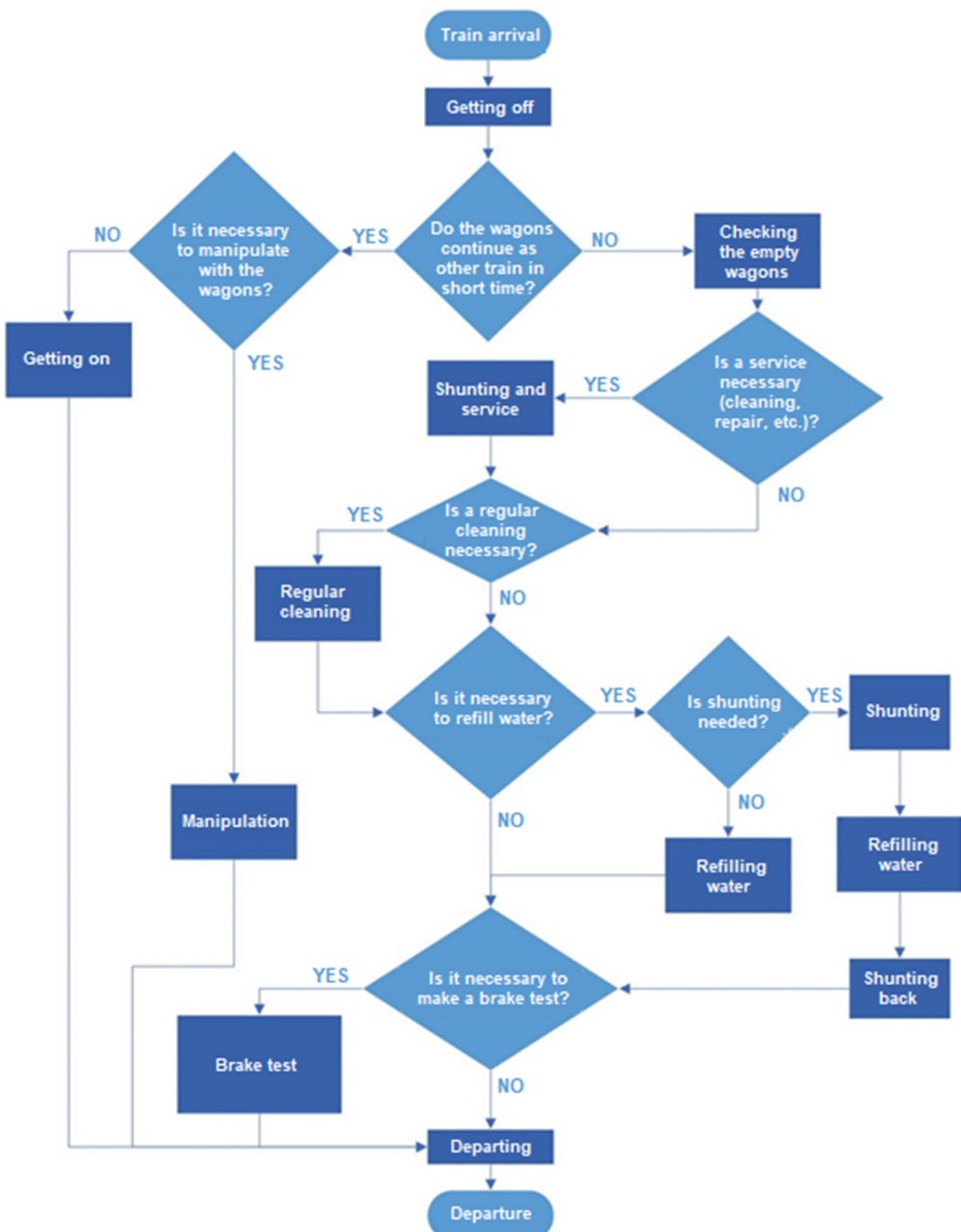

**Figure 7.** Technological operations from the arrival of the wagon set to its departure on the next train.

### 3.1. Design of Methodology for Automatic Creating of Circulation and Shiftplans

After a certain transformation, the principles applicable to the circulation of railway vehicles can also be applied to the shifts of operating staff, which will also be shown in specific cases. To determine the number of train sets that are needed to fulfil the timetable, i.e., the departure of each connection, which is listed in it, the circulation of sets is used.

The term circulation of sets means the sequence of individual connections in time, which determines the coverage of the given connections by sets (essentials). The circulation of sets is divided into individual days. It shows the sequence of connections to which a given set passes on a given day and also the sequence of individual days. During the circulation of the sets, all necessary technological tasks are taken into account, such as loading the fuel, operational cleaning, water replenishment, or emptying of fecal tanks. Set circulation is a basic tool for calculating the running of sets, for a period of one day, week, month, and year. However, the running values of the sets are always

calculated as an average, as the individual sets travel different amount of kilometers per day, but while maintaining the circulation, it is ensured that after a certain period, i.e., the total number of turning days, all sets have the same number of kilometers [46]. Circulation of sets must be handled separately for each day subject to timetable restrictions. It is also necessary to ensure a smooth transition of sets between individual days, even if it is switched between several circulations, which are a subject to restrictions [47]. The circulation of sets also shows whether the set is running individually or if two sets are running together on one train.

When compiling fixed circulations, the prescribed maintenance of rolling stock is considered. The combination of different types of railway vehicles in one circulation group is not allowed. An exception is a locomotive with the same or similar traction properties. When compiling the circulation of railway vehicles, it is necessary to have available technological procedures of work in the relevant railway station (e.g., technological times of travel to and from the train, or local shift, etc.). The planner submits a form in which he fills he circulation of the locomotive according to the train numbers for the relevant station, whose locomotive boards the trains, departs from the trains, or handle trains, a written request for the submission of technological journey times to and from the train, for each case of boarding or disembarking at the relevant station [48,49]. The time before the departure of the train must include the running time to the train, i.e., own ride of a rail vehicle from the border of the station or from the place of parking in the station, further possible shunting, time of preheating the train from the locomotive, or downtime at the station. Similarly, the time after the arrival of the train must include the indicated technological times.

All trains are incorporated into the circulation of rolling stock. Furthermore, all other regular services are incorporated into the circulation, such as spills, pushing, station shuntings, etc. The operational treatment, in the case of motorized locomotive, is also the loading of fuels, and in the case of electric and motorized wagons and units, the operational cleaning will also be incorporated into the circulation and marked. It is forbidden to incorporate operational treatment into the circulation of rolling stock in the event that this would increase the total need of vehicles [50]. In this case, the in-service treatment is performed by replacing the vehicle. If several rolling stocks are included in the train at the same time, the planner is responsible for the correct incorporation of the order of the rolling stock in the train. During the assembly of circulations, the requirements of the station for the times of (dis)connecting of vehicles to/from trains must be respected. Any case where the time between the arrival of the train at the station and the departure of the train is equal to or longer than the sum of the specified technological times before and after, the train is considered to be a fully compliant [51].

The issue of circulations of railway vehicles as well as the shifts of operating employees can be solved by various mathematical methods, but the mostly used solution of this issue is by using the transformation to the so-called "assignment problem". The assignment problem is defined as the task of optimally assigning elements from a set containing n elements to elements from an equally numerous other set. We assume that all values of the coefficients of the right sides $a_i; i = 1, \cdots, n$ and $b_j; j = 1, \cdots, n$ are equal to 1. The matrix $X$ consists of variables $x_{ijs}$, whose value is equal to 1, if the *i-th* element from the first set is assigned the *j-th* element from the second set, otherwise $x_{ij} = 0$. The coefficients $d_{ij}$ can be interpreted as time losses from the achieved assignments. Based on the above, we can formulate an assignment problem in the form:

$$\min z(x) = \sum_{i=1}^{n} \sum_{j=1}^{n} d_{ij} x_{ij} \tag{1}$$

$$\sum_{i=1}^{n} x_{ij} = 1 \ j = 1, 2, \ldots, n$$

$$\sum_{j=1}^{n} x_{ij} = 1 \ i = 1, 2, \ldots, n$$

$$x_{ij} \in \{0,1\} i = 1, \cdots, n; j = 1, \cdots, n$$

Due to the strong degeneracy, special methods have been developed for such tasks, such as the Hungarian method, the name of which is derived from the fact that the principle of the method is based on the theorem of the Hungarian mathematicians König and Egerváry [46,47].

Appropriate compilation of shifts in the requirements of rail passenger transport is an important task of transport technology. The aim of the set is to minimize the required number of requisites and ensure their maximum use. In doing so, many restrictive conditions must be met. By solving the assignment problem, we want to minimize time losses which represent differences between arrivals and departures of trains at a given station where the vehicles transfer to another train. Minimization of time losses will lead to a solution where a different train will be assigned to each train. This assignment represents the passage of the set between these trains [48].

Various software tools can be used to solve the assignment task. For example, MS Excel itself can solve the assignment problem as a task of linear programming, but only with a limited number of elements of the input matrix, so for a more complex solution it is necessary to use more advanced software tools [49,50].

The application of theoretical knowledge can be illustrated by a specific example, where a circulation of diesel engine units of the 813–913 series will be created on the transport route Košice–Moldava nad Bodvou mesto according to the train schedule 2017/2018, as can be seen in Table 1.

**Table 1.** Overview of trains on the line Košice–MnB mesto in years 2017/2018.

| Train Number | Departure | | Arrival | |
|---|---|---|---|---|
| 6400 | Košice | 4:58 | 5:38 | MnB mesto |
| 6401 | MnB mesto | 4:36 | 5:17 | Košice |
| 6402 | Košice | 6:35 | 7:23 | MnB mesto |
| 6403 | MnB mesto | 6:27 | 7:08 | Košice |
| 6404 | Košice | 10:11 | 10:54 | MnB mesto |
| 6405 | MnB mesto | 9:02 | 9:43 | Košice |
| 6406 | Košice | 14:11 | 14:54 | MnB mesto |
| 6407 | MnB mesto | 13:02 | 13:43 | Košice |
| 6408 | Košice | 16:11 | 16:51 | MnB mesto |
| 6409 | MnB mesto | 15:02 | 15:43 | Košice |
| 6410 | Košice | 18:25 | 19:05 | MnB mesto |
| 6411 | MnB mesto | 17:02 | 17:43 | Košice |
| 6412 | Košice | 22:25 | 23:05 | MnB mesto |
| 6413 | MnB mesto | 20:36 | 21:17 | Košice |

It is clear from the above timetable that according to the timeTable 2017/2018, there are 14 connections operating on the Košice–Moldava nad Bodvou transport route, 7 in the even direction and 7 in the odd direction. When creating the circulation of sets, it is considered that each train is assigned exactly one diesel engine unit of the 813–913 series. Solving this task would be empirically demanding, so the Solver add-on is used as a proven tool for assigning sets to trains.

An important prerequisite for the solution is the precise determination of the input data. In this case, it is a matter of determining the exact time difference between the arrival of the connection to the station and the departure of the connection from the same station. Since the trains are numbered according to the direction of travel, it could simply be said that it is a matter of determining the time differences between even and odd trains, more precisely between their arrivals and departures at the same station.

Table 2 shows the time differences between even and odd trains in minutes. The trains are numbered from 6400 to 612. For better clarity of the table, we use only the two last numbers (00–13). The train numbers are in the first line and first column, while the time differences are in the rest of the

table. After entering these input data, it is possible to start solving the assignment problem using the Solver add-on. The aim is to find such a combination of transitions that there are "time losses"—the differences between the arrival and departure of trains are minimal. It is necessary to put exactly one requisite on each joint—a diesel engine unit of the 813–913 series. The next step will be to arrange the results so that the circulation of the vehicles is created, in which the emphasis will be on the greatest possible efficiency of their use, which means to find the minimum number of vehicles that will be deployed on a given connection.

**Table 2.** Differences between train arrivals and departures in minutes.

|      | 00   | 01   | 02   | 03   | 04   | 05   | 06   | 07   | 08   | 09   | 10   | 11   | 12   | 13   |
|------|------|------|------|------|------|------|------|------|------|------|------|------|------|------|
| 00   |      | 1378 |      | 49   |      | 204  |      | 444  |      | 564  |      | 684  |      | 898  |
| 01   | 1421 |      | 78   |      | 294  |      | 534  |      | 654  |      | 788  |      | 998  |      |
| 02   |      | 1273 |      | 1384 |      | 999  |      | 339  |      | 459  |      | 579  |      | 793  |
| 03   | 1310 |      | 1407 |      | 183  |      | 423  |      | 543  |      | 677  |      | 917  |      |
| 04   |      | 1062 |      | 1173 |      | 1328 |      | 128  |      | 248  |      | 368  |      | 582  |
| 05   | 1155 |      | 1252 |      | 28   |      | 268  |      | 388  |      | 522  |      | 762  |      |
| 06   |      | 822  |      | 933  |      | 1088 |      | 1328 |      | 8    |      | 128  |      | 342  |
| 07   | 915  |      | 1012 |      | 1228 |      | 28   |      | 148  |      | 282  |      | 522  |      |
| 08   |      | 705  |      | 816  |      | 971  |      | 1211 |      | 1331 |      | 11   |      | 225  |
| 09   | 795  |      | 892  |      | 1108 |      | 1348 |      | 28   |      | 162  |      | 402  |      |
| 10   |      | 571  |      | 682  |      | 837  |      | 1077 |      | 1197 |      | 1317 |      | 91   |
| 11   | 675  |      | 772  |      | 988  |      | 1228 |      | 1348 |      | 42   |      | 282  |      |
| 12   |      | 331  |      | 442  |      | 597  |      | 837  |      | 957  |      | 1077 |      | 1291 |
| 13   | 461  |      | 558  |      | 774  |      | 1014 |      | 1134 |      | 995  |      | 68   |      |

The Solver add-on displays one optimal solution, but the task usually has several alternative solutions, in which case it is then necessary to take into account the relevant technological and economic aspects that affect the circulation of the vehicles—especially the efficiency of their use. The optimal assignments for the individual trains are given in Table 3 (marked with number 1).

**Table 3.** Solution of the assignment task on the session Košice–MB mesto.

|      | 00 | 01 | 02 | 03 | 04 | 05 | 06 | 07 | 08 | 09 | 10 | 11 | 12 | 13 |
|------|----|----|----|----|----|----|----|----|----|----|----|----|----|----|
| 00   | 0  | 0  | 0  | 1  | 0  | 0  | 0  | 0  | 0  | 0  | 0  | 0  | 0  | 0  |
| 01   | 0  | 0  | 1  | 0  | 0  | 0  | 0  | 0  | 0  | 0  | 0  | 0  | 0  | 0  |
| 02   | 0  | 0  | 0  | 0  | 0  | 1  | 0  | 0  | 0  | 0  | 0  | 0  | 0  | 0  |
| 03   | 0  | 0  | 0  | 0  | 0  | 0  | 1  | 0  | 0  | 0  | 0  | 0  | 0  | 0  |
| 04   | 0  | 0  | 0  | 0  | 0  | 0  | 0  | 1  | 0  | 0  | 0  | 0  | 0  | 0  |
| 05   | 0  | 0  | 0  | 0  | 1  | 0  | 0  | 0  | 0  | 0  | 0  | 0  | 0  | 0  |
| 06   | 0  | 0  | 0  | 0  | 0  | 0  | 0  | 0  | 0  | 1  | 0  | 0  | 0  | 0  |
| 07   | 1  | 0  | 0  | 0  | 0  | 0  | 0  | 0  | 0  | 0  | 0  | 0  | 0  | 0  |
| 08   | 0  | 0  | 0  | 0  | 0  | 0  | 0  | 0  | 0  | 0  | 0  | 1  | 0  | 0  |
| 09   | 0  | 0  | 0  | 0  | 0  | 0  | 0  | 0  | 1  | 0  | 0  | 0  | 0  | 0  |
| 10   | 0  | 0  | 0  | 0  | 0  | 0  | 0  | 0  | 0  | 0  | 0  | 0  | 0  | 1  |
| 11   | 0  | 0  | 0  | 0  | 0  | 0  | 0  | 0  | 0  | 0  | 1  | 0  | 0  | 0  |
| 12   | 0  | 1  | 0  | 0  | 0  | 0  | 0  | 0  | 0  | 0  | 0  | 0  | 0  | 0  |
| 13   | 0  | 0  | 0  | 0  | 0  | 0  | 0  | 0  | 0  | 0  | 0  | 0  | 1  | 0  |

From the above solution of the assignment task, it is necessary to compile a sequence of connections regarding the efficient use of vehicles or units. All connections must be included in the shift. The resulting circulation with a sequence of trains is shown in Table 4.

**Table 4.** The resulting circulation.

| Unit | Day | |
| --- | --- | --- |
| | **1** | **2** |
| 1. unit | 00 → 03 → 06 → 09 → 08 → 11 → 10 → 13 → 12 → 01 | 01 → 02 → 05 → 04 → 07 → 00 |
| 2. unit | 01 → 02 → 05 → 04 → 07 → 00 | 00 → 03 → 06 → 09 → 08 → 11 → 10 → 13 → 12 → 01 |

At least two train units are required to cover all connections according to the timetable. In practice, however, more than one unit is deployed on some trains, mainly due to the increase of transport capacity. The operation of some trains with more than one set can be mathematically defined so that the given trains (connections) and their arrivals and departures are given in the model example as many times as it is necessary to deploy them. In real operation, it is also necessary to take into account the technological processes with the vehicles, especially their repairs and maintenance. Operational maintenance of the set, which is to be performed at a specified time, must be part of the circulation of one of the unit and this time can therefore be deducted from the time lost between selected connections, or write operational maintenance as a special connection whose departure time will be start and arrival maintenance completion time.

*3.2. Draft Recommendations and Measures for the Carrier*

By solving the circulation of vehicles on the submitted timetable in the proposed concept of the methodology, it was proven that the input parameters (train timetable) have a great influence on the outputs (circulation of sets and train staff rotations). It is not possible to create quality circulation and shiftplan if the trains are run unevenly and during the day there is a large number of connections and disconnections in the individual stations and the sets are subsequently shut down or sent to the depot.

The process of automating the creation of circulations, but also shiftplans, is highly dependent on the input parameters [51]. The proposed methodology will always find the optimal solution, i.e., the best possible circulation, but it is important what input data are used i.e., what trains are given for the solution. Therefore, it is important to adhere to the following principles:

- Define the partial elements that will be rationalized—that is, to create circulations and shifplans along defined and logical parts, such as according to railway lines, transport routes, or regions. It is not appropriate to create circuits and shifts in such a way that we will consider only a limited number of selected trains from different lines. Such a solution will not be optimal from a complex point of view. It is, therefore, always necessary to insert all trains there according to the above proposals (lines, routes, regions).
- It is necessary to deploy only one type of railway vehicle (staff) on a defined element (lines, routes, regions). For example, on the selected line, locomotive of the same series should be deployed on all trains, not to combine them with each other. To increase the transport capacity in specific trains, it is necessary to add another locomotive but again of the same type. This solution is also suitable in terms of quality from the point of view of passengers, who will always know that only a given type of locomotive with the given parameters operates on a given route.
- From a technological point of view, it is much more advantageous to use complete units or return ones (push-pull) and with conventional sets (locomotive + wagons) it is advisable to use a control wagon at the end of this set so that the locomotive does not overflow when changing train direction, which has a negative impact on technological times. Thanks to this, the costs associated with the shift are also saved.
- Make circulations and shifts separately for each time and date limit. At the same time, try to reduce the number of these restrictions to avoid confusion. For example, it would be appropriate to have restrictions only on working days and weekends, with circulations and shifts being prepared separately for working days and weekends according to the timetable. It is ideal to run as many trains as possible without restrictions (daily).

- Last but not least, it is necessary to predispose quality circulations of railway vehicles to ensuring their quality maintenance in order to prevent failures. In the case of shifts of operational employees, it is necessary to positively motivate these employees to perform their service and give them an adequate salary for this performance.

Recommendations Concerning Train Staff

For the preparatory work before the departure of the train from the departure station, it would be appropriate to consider the position of the "driver-preparer", whose job would be to comprehensively prepare the departure train [52,53]. In connection with the activities performed so far by the driver, it is possible to assume a more efficient use of the working time of the driver, where the technological act of taking over the train locomotive will be carried out "on the route". At the same time, this will save time but also eliminate the inefficient work performances of the driver. We propose to incorporate this recommendation into the catalogue of typical job positions, specifying which driver is also the driver-preparer.

Recommended activities for the proposed job position "driver-preparer" are linked to a set of activities from taking over the key from the locomotive in the depot/platform (in case of terminating train) to braking locomotive on the platform and handing over the keys to the train driver (or locking the locomotive in depot).

As far as the accompanying staff is concerned, in connection with the proposal of the methodology for the processing of train circulations, we have come to the conclusion that the organization of the operating staff centers (locomotive depots) does not meet the criteria for possible optimization. When creating a new timetable, it is appropriate to consider the relocation of these centers to the departure and turning stations of trains. This will save the necessary overhead travel for employees from the centers to perform work, minimize on-call time at workplaces, eliminate inefficient performances, and improve the working conditions of employees.

## 4. Discussion

It is necessary to approach the problem of optimizing circulation and shiftplans in a comprehensive way, and it is appropriate to base the solution on existing research focused on this issue [54]. The most important prerequisite, however, is a thorough analysis of the current state of the problem at home and abroad, not only from a scientific but also from a practical point of view [55]. It is necessary to summarize all previous knowledge in the areas of the legal framework, information technology, transport processes, and transport theory in connection with the issue of shifts. The set of all analyzed knowledge forms the input data for optimization.

The effectiveness of the optimization is directly dependent on the input data. These must be relevant and exact; thus, a thorough analysis is essential [56,57]. The aim of the design of the methodology for the creation of optimal train circulation is conditioned by the structure and quality of the input data. In contrast to previous research by other authors, the added value lies in the description of the possibilities of using known mathematical methods for rail passenger transport [57–60]. In practice, this problem is solved empirically, and in scientific publications, the applications of selected mathematical algorithms were applied only to urban public transport, which has different specifics than rail passenger transport [56,61,62]. For buses and trolleybuses, there are no changes of wagons or locomotives. The algorithms used must take into account all the specific characteristics of rail passenger transport, the definition of which is part of the description of the methodology for creating optimal shifts. In this way, the article combines the existing scientific knowledge of the issue with new possibilities of their use and in this way contributes to increasing the value of scientific research in rail passenger transport [63,64].

It is possible to approach the solution of this issue from various theoretical starting points. At present and in the environment, a significant increase in the use of modern information technologies can be seen in the field of transport, the main task of which is to simplify work activities [65–67].

The physical output of the solution to the problem could be, for example, software that would be able to create optimal circulations and shiftplans for the needs of the carrier based on the entered input data [62,68]. Nevertheless, the development of such software requires the methodology on the basis of which the software should work. Incentives for its further application can therefore be found in the information technology sector. Thanks to them, it would be possible to achieve automation of the creation of train essentials and at the same time be sure that the resulting rounds are optimal. Practical use is possible not only in the strategic, but also in the operational level of planning in the transport company. Train tours could thus process computers instead of people, which has its advantages and disadvantages.

The organization of railway transport is different from other means of transport in many fields. It can be also considered as a complex process that consist of partial operations and a lot of other factors. One of the most vital is the customer—not only as a final segment, but also as a judge for quality and source of incomes [66,69,70]. From his point of view, one of the most important measures is the timetable. However, creating a timetable is only the beginning of the long process. For our research, a timetable was considered only as input data. It is also necessary to create more plans for a good train operation and success of the whole carrier, such as optimal circulation and shiftplans [70,71]. At present, in all cases, they are formed only empirically, which can result in an inefficient result, thus a big increase in costs. In the end, this can make profit shrink.

This hypothesis is substantiated by the fact, which happens very often, that a responsible employee (planner) creates a shiftplan that is then a subject to change. Sometimes, even the operation employees (engine drivers, conductors, etc.) suggest an improvement themselves, so it is clear that for one person, it is almost impossible to find the optimal solution without mathematical or other methods [72,73]. Regarding this, our research applicates the assignment problem to solve this problematic, as can be seen from the case study.

Another problematic fact is that necessary technological processes vary for almost each station, train, case, day, etc. Their length is nowadays only estimated or replaced by a "standard", which often does not reflect the reality [74,75]. Therefore, we suggest other methods, such as calculation on the basis of influence diagrams. It seems to be the ideal tool to take into account various conditions. On the other hand, it must be mentioned that for the exact result of these calculation, a huge amount of input information is needed. This condition can be met only after creating an enormous database with detail information about all the trains, stations, and staff. In today's operating conditions, something like this is hardly imaginable.

As mentioned in the background, the requirements of the carrier are antagonistic to the requirements of its employees [76,77]. Mathematical methods create an optimal circulation and shiftplan for a carrier, however, this may not be suitable for the employees. One could say, that it may lead to a demotivation, but the fact is, that the majority of operational railway staff is already demotivated nowadays [78,79]. This enormous problem can cause not only psychologic problems, but in the end a lack of the quality of the whole transport service, which means that very new idea must be implemented or at least tried. The carriers are already trying to improve the situation by hiring special managers and experts for human behavior and motivation. Another way can be improving the essence of the job—the shifts.

This is the reason that we present a software solution with an interface for the employees. It enables the staff to make a shift preference, which leads to the fact, that in the plan for a next time period e.g., a month, the system deploys as many preferred shifts to the particular employee. Regarding the shifts, it can be also possible to plan a day offs for the next period in a more sophisticated way than nowadays (calling). This is also helpful for the managers and planners. Another way of facilitation of the work is to digitalize the paper shift reports, which will save time as well as enable creation of digital data database, that can be used for trend monitoring as well as predictions.

Besides innovative solutions, we can also state several other recommendations to the carrier to improve the processes. Those were concluded from the research and relate also to the train staff and

creation of circulations and shiftplans. They do not outline a new methodology, but changes that may lead to better efficiency of the above-mentioned procedures. If implemented by the carrier, there may be even greater efficiency than simply using the methodology itself.

The scientific enrichment of the proposed methodology lies in the mathematical procedures applied in it and their application. Summarization and subsequent analysis of all theoretical knowledge is important in terms of the complexity of solving this problem, as in previous research the application of mathematical procedures in rail passenger transport did not occur [80,81]. In terms of use in practice, the proposed methodology can be used in the management of carriers in passenger transport at the strategic and operational level. This is because the circulations and shifts are planned in the long term, usually for the relevant train timetable, but may change operationally, for example due to closures or delays [82,83]. The methodology of creating optimal plans is designed to be flexible, which increases the number of possibilities of its use. It is directly dependent on the input data, so if they are correctly interpreted, it is possible to create an optimal circulation or shiftplan according to any conditions. It is primarily used to create plans for a longer period of time, but thanks to its flexibility, there is the possibility of operational changes. For example, if a train is delayed, it is possible to change the input "train arrival" to take into account the current delay and with this new input to create a new plan that will be optimal. The carrier thus streamlines its management from an operational as well as an economic point of view.

When considering the problematic, among other things, a thorough analysis of other similar procedures, such as abroad or with competing companies, is important for a successful design [84]. Nevertheless, computer processing and circulation is a completely new idea in this area. For this reason, it is difficult to base or compare research on other professional publications. All the research tasks are nowadays performed empirically as mentioned in the background [85,86].

On the other hand, there is a huge opportunity for research in the future. More work could be done to improve the methodology. Our system is based only on stable input data and factors. It would be more effective if there was a function of considering small changes in a few parameters that could lead to much better result. For example, a proposal of changing the place of depots, beginning of the shifts, type of wagons or locomotives, train routes, etc. The carrier would be able to simply calculate the costs for implementing such change and compare it to savings obtained. When thinking this way, we can say, that a software would generally help the management with making better decisions. Such "smart" changes could eventually be related also to the timetable as a basis; however, more analysis would be needed, and the results are unpredictable from today's point of view.

Another topic might be possible synergistic effects arising from all the above mentioned. The complex problem deals with an enormous number of variables, therefore there might be a lot of bigger improvements based only on a combination of few smaller changes. This is also a field where more analysis is needed.

The most challenging task is the IT background. Regarding the assignment problem and software designing, this is a research topic not only for the experts for transport, but also for the ones for information technologies [87]. There are a lot of possible solutions and inter-domain cooperation. Current computers allow a number of other methods that would be suitable for this area. In addition, the sector is experiencing tremendous growth.

As for improving the conditions of the employees, more research can be done too. The transport market is developing around the clock, which means that not only passengers' demands change, but also the staff's ones [88]. More case studies could help us to understand the railwaymen better and that could lead to another improvement. What is more, a comparison of the staff's behavior, demands, and opinions from different carriers or places could provide us an interesting sociological data.

As mentioned above, the issue of rail passenger transport is very extensive and at the same time very specific. In-depth analysis and solution of case studies, a large amount of different information, data, and parameters are considered, however some facts may be omitted, or some influence may be neglected. For this reason, it would be best to support theoretical knowledge through years of

experience in solving problems in the field. To improve the quality of research, its content was consulted with many experts from transport companies as well as from the scientific and academic community, which created the possibility of finding a consensus between theory and practice.

Solving vehicles' circulation and the train staff's shiftplans using selected mathematical methods is a necessary step to optimize them. Mathematical methods are the basis of information systems, for which the correct design and use it is important to know all the algorithms on which the information systems work. In order for the outputs of algorithms to be properly understood and interpreted, these information systems must provide users with correct and relevant solutions. It is a way in which decision-making processes can be simplified and improved in a transport undertaking, for example when assigning sets or train requirements to individual connections.

**Author Contributions:** Conceptualization, J.Č. and E.B.; data curation, J.Č. and M.P.H.; formal analysis, J.Č., and E.B.; funding acquisition, J.Č.; investigation, E.B. and M.P.H.; methodology, J.Č.; project administration, J.Č.; E.B., and M.P.H.; software, J.Č.; validation, J.Č. and E.B.; visualization, M.P.H.; writing—original draft, E.B. and M.P.H.; writing—review and editing, J.Č. All authors have read and agreed to the published version of the manuscript.

**Funding:** This research was supported by Scientific Grant Agency: The project P-101-0460/19 Railway Company Zeleznicna Spolocnost Slovensko; KEGA No. 014ŽU-4/2020, Six Sigma and progressive education of the quality management in the railway transport field of study in accordance with the requirements of railway undertakings, of the Ministry of Education, Science, Research and Sport of the Slovak Republic.

**Conflicts of Interest:** The authors declare no conflict of interest.

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
