# Peer review of "Information System and Technology Optimization as a Tool for Ensuring the Competitiveness of a Railway Undertaking—Case Study"

_sustainability, doi:10.3390/su12218915_

Round 1

Reviewer 1 Report

The authors did a good job.

Author Response

Dear reviewer,

thank you very much for your willingness and time. We are pleased that you find our research useful. We would like to add that the text has been changed according to another comments of reviewers, few literature sources were added into the introduction and the text has been re-proofread by the agency again. Figures are rotated according to reviewers' requests. All changes are in green colour in the text.

Thank you very much again.

Best regards,

Juraj Camaj

19 October 2020

Reviewer 2 Report

The paper concerns a new tool for optimizing the rail operations. The methodology is consistent and the information presented is new.

However, some revisions are necessary. 

Firstly, the authors have dealt with the information system KASO but no explanation at all is provided to it. It is mandatory that the authors provide at least a brief description of the IS KASO.

Moreover, figures 3 and 4 are not very readable, especially figure 3. I would suggest to turn figure 3 by 90° and to make it occupy the entire page.

The English is very good and very understandable, but there are a few typos to correct.

Author Response

Dear reviewer,

thank you very much for your willingness, time and fruitful comments and suggestions. We are pleased that you find our research useful. Please, here you can find our point-by-point response. We do hope it fulfil other requirements:

  • An explanation of the IS KASO has been provided.
  • Figures 3 and 4 have been changed and improved their readability.
  • All changes are in green colour in the text.
  • We would like to add that the text has been re-proofread by the agency again.
  • The text was also expanded to include a list of foreign literature.

Thank you very much again.

Best regards,

Juraj Camaj

20 October 2020

Reviewer 3 Report

This paper proposes a mathematical method to solve the train’s circulation and staff’s shift plans. The present method is information-based and has the potential to facilitate the decision-making process. Generally, this paper has highly scientific value and is very promising to reduce the cost of the railway operator. The reviewer recommends the publication of the paper by addressing the following comments.

The methodology and analysis presented in the article strongly rely on the accuracy and exactness of the input data. Please comment on how the to ensure the relevant and exact of the input data in the present model.

Apart from the traction rail vehicles, train drivers and train crews presented in paragraph 1, the maintenance of railway infrastructures such as track [1] and overhead line [2], mechanical [3] and electrical [4] components inside the train is also the area of importance to reduce the cost. Please consider enriching the literature review in the introduction.

[1] S. Iwnicki, Handbook of railway vehicle dynamics, CRC Press, 2006. https://doi.org/10.1201/9781420004892.

[2] Y. Song, Z. Liu, A. Rønnquist, P. Nåvik, Z. Liu, Contact Wire Irregularity Stochastics and Effect on High-Speed Railway Pantograph–Catenary Interactions, IEEE Trans. Instrum. Meas. 69 (2020) 8196–8206. https://doi.org/10.1109/TIM.2020.2987457.

[3] Z. Wang, Y. Song, Z. Yin, R. Wang, W. Zhang, Random Response Analysis of Axle-Box Bearing of a High-Speed Train Excited by Crosswinds and Track Irregularities, IEEE Trans. Veh. Technol. 68 (2019) 10607–10617. https://doi.org/10.1109/TVT.2019.2943376.

[4] Chen, Minwu, Yinyu Chen, and Mingchi Wei. "Modeling and control of a novel hybrid power quality compensation system for 25-kV electrified railway." Energies 12.17 (2019): 3303.

Please mind some editorial issues in the texts and figures, such as  

Please improve the quality of Figure 4 to make it more clear and readable.

In line 85, please modify the section title.

Author Response

Dear reviewer,

thank you very much for your willingness, time and fruitful comments and suggestions. We are pleased that you find our research useful. Please, here you can find our point-by-point response. We do hope it fulfil other requirements:

  • An explanation of the IS KASO has been provided.
  • Figures 3 and 4 have been changed and improved their readability.
  • All changes are in green colour in the text.
  • We would like to add that the text has been re-proofread by the agency again.
  • Regarding the inputs to the system - real inputs from railway operation are provided in the model. The inputs are from the carrier's sources, so relevance is ensured directly in cooperation with the railway undertaking.
  • The text was also expanded to include a list of foreign literature. Thank you for your contribution to the literature sources, we found them very interesting.
  • In line 85, the section title has been modified.

Thank you very much again.

Best regards,

Juraj Camaj

20 October 2020